# Early biological markers of post-acute sequelae of SARS-CoV-2 infection

Scott Lu[1,2,11], Michael J. Peluso [3,11], David V. Glidden[2], Michelle C. Davidson [4], Kara Lugtu[1], Jesus Pineda-Ramirez[1], Michel Tassetto[5], Miguel Garcia-Knight[5,6], Amethyst Zhang[5], Sarah A. Goldberg [2], Jessica Y. Chen [2], Maya Fortes-Cobby[1], Sara Park[1], Ana Martinez[1], Matthew So[1], Aidan Donovan[3], Badri Viswanathan [2], Rebecca Hoh[3], Kevin Donohue [4], David R. McIlwain[7], Brice Gaudiliere [7], Khamal Anglin[1], Brandon C. Yee [8], Ahmed Chenna[8], John W. Winslow[8], Christos J. Petropoulos [8], Steven G. Deeks [3], Melissa Briggs-Hagen[9], Raul Andino [5], Claire M. Midgley [9], Jeffrey N. Martin [2], Sharon Saydah [9,12] & J. Daniel Kelly [1,2,4,10,12] ✉

To understand the roles of acute-phase viral dynamics and host immune responses in post-acute sequelae of SARS-CoV-2 infection (PASC), we enrolled 136 participants within 5 days of their first positive SARS-CoV-2 real-time PCR test. Participants self-collected up to 21 nasal specimens within the first 28 days post-symptom onset; interviewer-administered questionnaires and blood samples were collected at enrollment, days 9, 14, 21, 28, and month 4 and 8 post-symptom onset. Defining PASC as the presence of any COVID-associated symptom at their 4-month visit, we compared viral markers (quantity and duration of nasal viral RNA load, infectious viral load, and plasma N-antigen level) and host immune markers (IL-6, IL-10, TNF-α, IFN-α, IFN-γ, MCP, IP-10, and Spike IgG) over the acute period. Compared to those who fully recovered, those reporting PASC demonstrated significantly higher maximum levels of SARS-CoV-2 RNA and N-antigen, burden of RNA and infectious viral shedding, and lower Spike-specific IgG levels within 9 days post-illness onset. No significant differences were identified among a panel of host immune markers. Our results suggest early viral dynamics and the associated host immune responses play a role in the pathogenesis of PASC, highlighting the importance of understanding early biological markers in the natural history of PASC.

Post-acute sequelae of SARS-CoV-2 infection (PASC) include ongoing symptoms in the months following acute COVID-19. Estimates of the occurrence of PASC can vary, but as of 2024, estimates suggest that 6.9% of U.S. adults have ever had PASC[1,2]. Despite a growing understanding of its epidemiology and natural history, the pathogenesis of PASC remains incompletely understood[3]. Multiple mechanisms that could contribute to this condition are now under investigation[3,4].

Viral antigen persistence and immune dysregulation are two mechanisms that might drive PASC[5–10]. For example, recent work has suggested that a high proportion of individuals with PASC demonstrate detectable SARS-CoV-2 antigen in blood plasma during the post-acute phase[5,7,11], subgenomic RNA has been identified in widespread tissue sites at autopsy for up to 6 months post-COVID[6], and RNA has been found in gut and other tissues in living individuals[12,13]. Furthermore, studies comparing individuals with PASC with those who report

**Fig. 1 | Flow diagram of participants evaluated for post-acute sequelae of SARS-CoV-2 infection (PASC) at month 4.** Flow diagram showing number of enrolled participants, COVID-19 and uninfected participants, and final number of participants who completed follow-up.

complete recovery from SARS-CoV-2 infection have demonstrated increased levels of certain inflammatory markers including IL-6, TNF-α, and IL-1B, among others, for at least a year following infection[8–10].

Biological samples to assess predictors of PASC from the acute and early post-acute phases are limited. Most studies from early infection have focused on shorter-term outcomes in hospitalized individuals[14,15], although more recently some studies have suggested that prolonged viral clearance[16,17] or distinct early immune signatures[18] could be associated with PASC. While informative, these studies have been limited by relatively small study populations, an abundance of hospitalized individuals, and/or infrequent biospecimen collection timepoints.

Here, we leverage a household-based cohort of individuals intensively sampled during the acute phase of SARS-CoV-2 infection and followed prospectively to assess for early biological determinants of PASC by comparing those known to develop PASC to those who recovered. We hypothesized that selected biological processes measured during early infection play a role in the pathogenesis of persistent symptomatology following the acute period of illness.

## Results

### Study participants
From September 2020 to May 2022, we enrolled 136 SARS-CoV-2 infected participants during the acute phase of their illness (Fig. 1). Among these participants, 104 completed at least one post-acute visit and contributed at least one nasal sample for infectious viral and RNA testing (87% of participants had 10 or more nasal specimens). These participants were all enrolled during their first confirmed case of COVID-19. Among these 104 participants, 80 participants also contributed blood for viral N-antigen and inflammatory marker testing.

Participants had a median age of 35.5 years (IQR: 27 to 44) (Table 1), were 51% (n = 53) self-identified females and were generally healthy with most (77%) reporting no pre-existing comorbid medical

conditions; the most common comorbidities were lung disease (14%), hypertension (10%), and diabetes (5%). The majority of participants (93%) were infected with pre-Omicron strains of SARS-CoV-2 and most (65%) had not received a SARS-CoV-2 vaccine prior to their infection. Over the course of their acute illness, 96 of 104 (92%) participants reported the presence of at least one symptom with a median of 9 symptoms (IQR: 4 to 13). The most common acute symptoms were fatigue (76%), rhinorrhea (74%), cough (71%), headache (54%), and sore throat (46%). See Supplemental Tables 1–2 for characteristics of the subgroup of participants who had viral N-antigen and cytokine testing. These participants had similar characteristics as those in the overall cohort, except that all participants in the subgroup were infected with a pre-Omicron variant.

### Description of PASC
Among 104 SARS-CoV-2 infected participants with a post-acute follow-up visit, 32 (31%) reported new, worsened or persistent symptoms since the time of SARS-CoV-2 infection, consistent with PASC. We did not identify a difference between those with and without PASC in terms of vaccination status (32% vs 36%). All participants with PASC had been symptomatic during acute illness with a median of 12 (IQR: 9 to 14) symptoms noted; among those without PASC the median number of symptoms in the acute period was 7 (IQR: 3 to 12). No participants reported any reinfection at the time of PASC status ascertainment.

Those meeting criteria for PASC reported a median of 2 (IQR: 1 to 5) symptoms during the post-acute period 2 to 6 months after SARS-CoV-2 infection. The most commonly reported PASC symptoms were trouble with concentration/memory (44%), trouble with sleep (28%), fatigue (25%), and rhinorrhea (25%) (Supplemental Table 3).

### Associations of virologic factors with PASC
We compared the magnitude, decay rates, and duration of nasal RNA and infectious viral shedding and plasma N-antigen levels between participants who did and did not develop PASC (Table 2). Compared to those who did not develop PASC, those who developed PASC had higher maximum nasal RNA viral load and plasma N-antigen levels (Table 2; Fig. 2a). After adjustment, plasma N-antigen at day 5 (p = 0.01) and nasal maximum RNA viral loads demonstrated statistical significance (Maximum RNA N viral load: p = 0.02; Maximum RNA E viral load: p = 0.03). The development of PASC was associated with a higher proportion of nasal RNA positivity (p < 0.001) and infectious viral shedding (p = 0.02; Fig. 3) during the 28 days following symptom onset. Viral RNA decay rates did not significantly differ by PASC status. In sensitivity analyzes with and without correction for multiple comparisons, we found similar results (Supplemental Table 4).

### Associations of host immune factors with PASC
Anti-RBD IgG levels among unvaccinated participants exhibited an increasing trajectory during the 28 days following symptom onset. Notably, those who developed PASC exhibited lower levels of Spike IgG at day 5 (p = 0.03) and day 9 (p = 0.03) post-symptom onset compared to those who did not develop PASC (Fig. 2b). These differences attenuated after day 14 and both groups had similar Spike IgG levels by day 28.

Inflammatory biomarkers exhibited a decreasing trajectory during the 28 days following symptom onset (Fig. 4). Those who developed PASC had non-significant trends toward higher initial levels of MCP, IL-10, IFN-α, and IFN-γ that attenuated over the observation period. Overall, there were no strong associations between starting levels or trajectories of inflammatory markers (IL-6, IL-10, TNF-α, MCP, IP-10, IFN-α, and TFN-γ) during the acute phase and the later development of PASC within this outpatient cohort.

## Table 1 | Baseline characteristics of the cohort

|  | Total |
|---|---|
|  | *N* = 104 |
| Age (years), median (IQR) | 35.5 (27 to 43.5) |
| Sex, *n* (%) |  |
| Female | 53 (51%) |
| Male | 51 (49%) |
| Race/Ethnicity, *n* (%) |  |
| Hispanic or Latino Ethnicity | 21 (21%) |
| White | 59 (58%) |
| Black or African American | 4 (4%) |
| Asian | 14 (14%) |
| Prefer not to answer | 4 (4%) |
| Vaccination status, *n* (%) |  |
| Unvaccinated | 68 (65%) |
| Fully vaccinated[a] | 36 (35%) |
| BMI, *n* (%) |  |
| <25 | 51 (51%) |
| 25 to 29.9 | 27 (27%) |
| >=30 | 22 (22%) |
| Education, *n* (%) |  |
| Less than HS | 6 (7%) |
| At least some HS | 9 (11%) |
| At least some college | 42 (49%) |
| At least some graduate school | 28 (33%) |
| Variant[b], *n* (%) |  |
| Pre-Delta | 61 (59%) |
| Delta | 36 (35%) |
| Omicron | 7 (7%) |
| Any comorbidity[c], *n* (%) |  |
| No | 80 (77%) |
| Yes | 24 (23%) |
| Number of symptoms in acute illness, median (IQR) | 9 (4 to 13) |
| Maximum RNA viral load (N viral target), median (IQR) [d] | 9.2 (7.0 to 10.5) |
| Maximum RNA viral load (E viral target), median (IQR) | 8.7 (6.8 to 10.0) |
| Duration of RNA viral shedding in days, median (IQR) | 9 (6 to 12) |
| Maximum infectious viral load, median (IQR)[e] | 8.2 (6.0 to 9.8) |
| Duration of infectious viral shedding in days, median (IQR) | 5 (0 to 7) |

[a]Fully vaccinated was defined as completion of primary vaccination series greater than 2 weeks before enrollment.
[b]Variant status was defined by viral sequencing results for 70 (67%) participants and by calendar time for the remaining 34 (33%) participants.
[c]Any comorbidity was defined from the following list: history of autoimmune disease, cancer in the past 2 years, diabetes, HIV/AIDS, heart disease, hypertension, lung disease, or kidney disease.
[d]RNA viral load measurements reported as log-transformed copies/mL.
[e]Among participants with a viral measurement, there were 52 (50%) who had maximum infectious viral load assessed, measured in plaque forming units.
*BMI* body mass index, *HS* high school.

## Discussion

In this non-hospitalized, household-based cohort of outpatients followed prospectively from the time of SARS-CoV-2 symptom onset through the post-acute period, we identified several early biological determinants of PASC. Specifically, we found a relationship between early viral dynamics, antibody responses, and the later development of PASC. These findings suggest that an individual's ability to control viral replication may play an important role in recovery from SARS-CoV-2 infection and that the dynamics of the immune responses during the acute and early post-acute phase may determine who goes on to experience post-acute symptoms. Our observations also provide additional rationale for the evaluation of early antiviral therapy as one potential approach for mitigating the development of PASC.

The etiology of PASC remains incompletely understood and there are no specific treatments available beyond symptom management. To date, most biological assessments of PASC have focused on the post-acute phase of infection[5,7–10,19–25]. Only a few studies have assessed the relationship between biomarkers in the acute phase of SARS-CoV-2 infection and the later development of PASC[16–18,26–28]. Such efforts have identified the presence of SARS-CoV-2 RNA in blood at the time of diagnosis[26], prolonged duration of viral RNA shedding from the upper respiratory tract[16], and higher respiratory burden of SARS-CoV-2[17] as correlates of PASC. Overall, these efforts to understand the relationship between acute-phase biology and post-acute symptomatology have been limited by the considerable challenges related to collecting specimens during the earliest days following COVID-19 symptom onset. As a result, many studies have been limited to individuals hospitalized with COVID-19, who do not represent most individuals experiencing PASC, the majority of whom were never hospitalized[29].

Using orthogonal virologic assessments, we observed a relationship between the early viral burden and the development of PASC in a non-hospitalized population. Those who went on to develop PASC had higher maximum levels of nasal SARS-CoV-2 RNA and plasma N-antigen during acute infection, suggesting that higher peak viral load is related to the later development of PASC. Furthermore, we found that those who later developed PASC exhibited a greater burden of RNA and infectious viral shedding over time and had a less robust humoral immune response in the first 9 days following infection, potentially tied to more sluggish viral clearance. Taken together, these findings suggest that the total amount of virus present, the immunologic response that results in control of that virus, and the efficiency of clearing replicating virus might all drive the development of post-acute symptoms.

A number of demographic and clinical risk factors for PASC have been identified. While prior SARS-CoV-2 vaccination appears to be protective against the development of PASC[30], it remains unclear whether antiviral treatment during the acute phase of illness might reduce the incidence of this condition. Two studies suggest a benefit among those meeting the criteria for antiviral therapy[31,32], but other evaluations have not shown an effect[33,34]. Our findings suggest that interventions that alter the duration of RNA and/or infectious virus shedding and the associated immune response during the acute phase might have the potential to affect the development of PASC. These could include antivirals (e.g., protease or RNA polymerase inhibitors) to reduce viral replication and/or monoclonal antibodies or therapeutic vaccination to enhance the immune response needed to neutralize the virus. Our observation of a delayed anti-RBD antibody response demonstrates some evidence towards the role of host immune response in the pathologic mechanisms that lead to PASC. As our comparison of immune responses was limited to a strictly unvaccinated sub-group, further assessment of the impact of interventions during the acute phase of infection is needed in a vaccinated population.

Acute COVID-19 is a highly inflammatory condition[14,15]. While prior cross-sectional and longitudinal analyzes in the post-acute phase have demonstrated that PASC is associated with differences in markers of immune activation[8–10,19,21,26], we did not identify significant differences in levels of these markers during the acute phase between those who did and did not go on to develop PASC. There are several possible explanations for this. First, our cohort was likely to be more homogeneous in terms of disease severity than those included in other

**Table 2 | Association of virologic factors with post-acute sequelae of SARS-CoV-2 infection (PASC) (N = 104)**

|  | Non-PASC N = 72, median (IQR) | PASC N = 32, median (IQR) | Incidence Ratios[c] (95% CI) | p-value |
|---|---|---|---|---|
| Maximum RNA N viral load[a] | 6.7 (4.8 to 8.3) | 8.3 (7.2 to 9.5) | 1.21 (1.03 to 1.43) | p = 0.02 |
| Maximum RNA E viral load | 6.3 (4.7 to 7.9) | 7.6 (6.7 to 8.6) | 1.18 (1.01 to 1.39) | p = 0.03 |
| Maximum infectious viral load[a,b] | 5.3 (3.0 to 6.1) | 5.2 (4.2 to 5.9) | 1.08 (0.90 to 1.32) | p = 0.43 |
| Rate of RNA N viral decay | −0.3 (−0.5 to −0.1) | −0.4 (−0.5 to −0.2) | 0.66 (0.26 to 1.70) | p = 0.39 |
| Rate of RNA E viral decay | −0.4 (−0.6 to −0.2) | −0.5 (−0.7 to −0.3) | 0.73 (0.46 to 1.16) | p = 0.19 |

[a]Viral load measurements were log-transformed copies/mL.
[b]Among 104 participants with a viral measurement, there were 53 (25 PASC, 28 non-PASC) who had maximum infectious viral load assessed.
[c]Incidence ratios calculated with delta-method standard errors using generalized estimating equations controlling for sex, age, vaccination status, and SARS-CoV-2 variant.

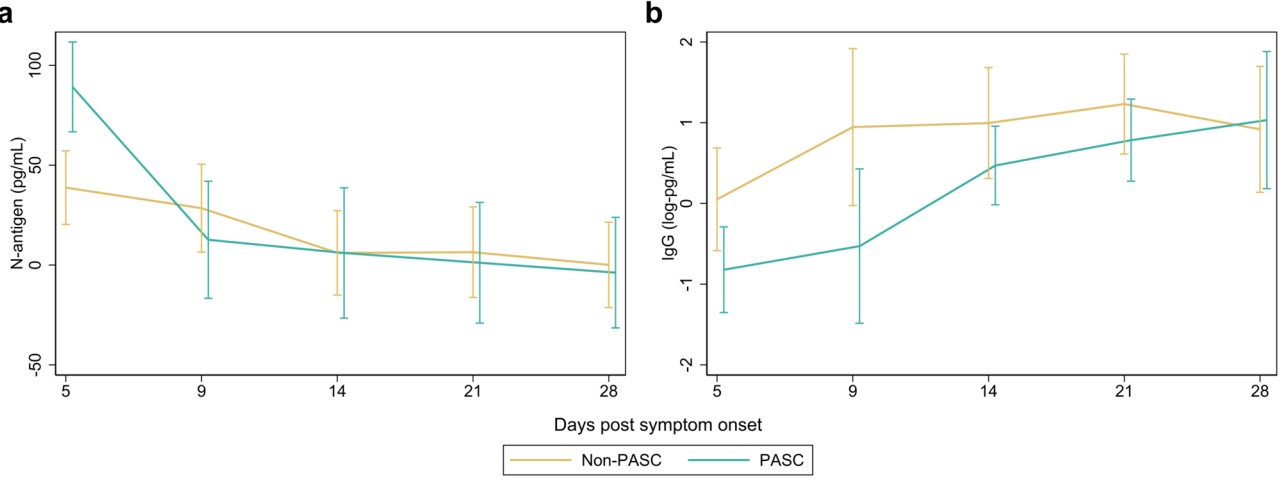

**Fig. 2 | Comparison of plasma N-antigen and IgG Spike antibody levels among those with and without post-acute sequelae of SARS-CoV-2 (PASC) over a 28-day period after symptom onset. a, b** Estimated mean values and 95% confidence intervals are plotted for biological specimen among PASC and non-PASC groups at days 5, 9, 14, 21, and 28 for N-antigen (**a**) and IgG (**b**) using generalized estimating equations fit with independent correlation, identity linkage, and Gaussian distribution. Covariates included sex, age, vaccination status, and SARS-CoV-2 variant. Statistics are derived from 80 participants with N-antigen measurements and 48 unvaccinated participants with IgG measurements. Statistical significance was assessed using two-tailed tests. Statistically significant distributions include N-Ag at day 5 (*p* = 0.01), IgG spike antibody at day 5 (*p* = 0.04), and IgG spike antibody at day 9 (*p* = 0.03).

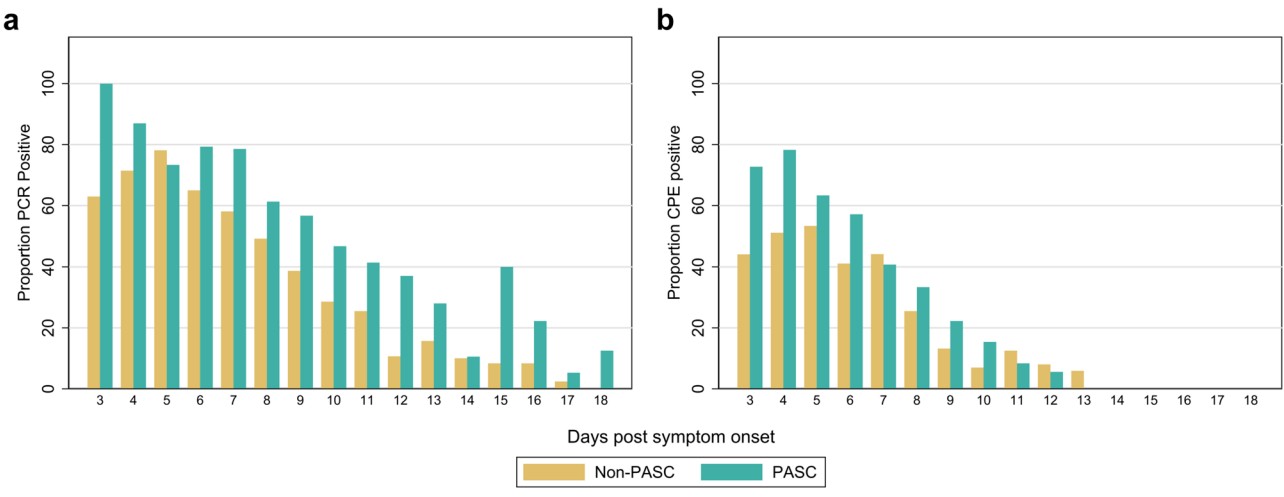

**Fig. 3 | Proportion of RNA and infectious viral shedding for each day after symptom onset among those with and without PASC. a, b** Bar graphs demonstrating proportion of nasal samples positive for (**a**) viral shedding and (**b**) infectious virus by day post-symptom onset from day 3 to day 18 among 32 PASC and 72 non-PASC participants. Two-sided pooled logistic regression demonstrated statistically significant difference in the proportion of positive viral shedding (*p* < 0.001) and infectious virus (*p* = 0.02).

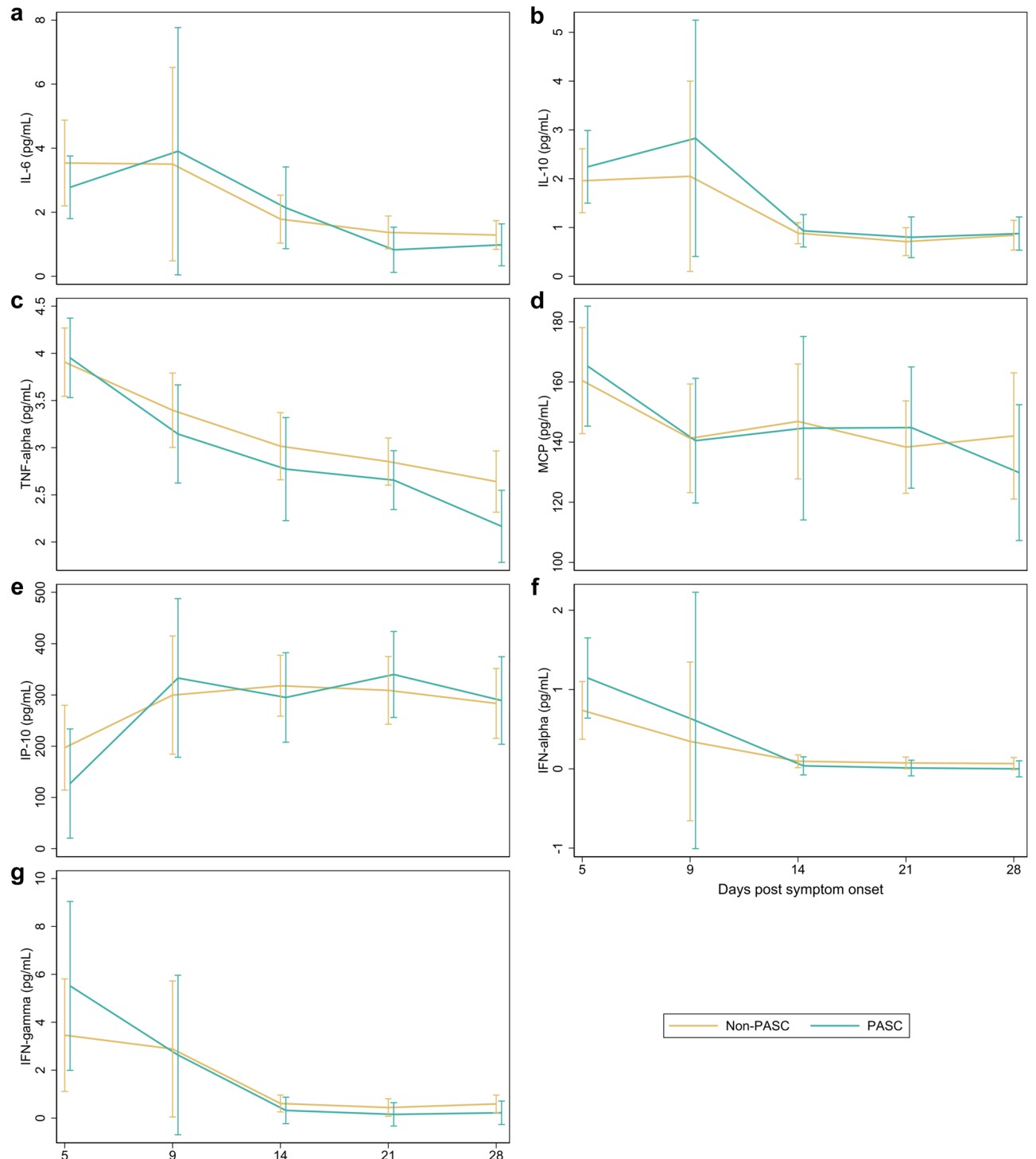

**Fig. 4 | Inflammatory marker levels among those with and without post-acute sequelae of SARS-CoV-2 infection (PASC) over a 28-day period after symptom onset. a–g** Estimated mean values and 95% confidence intervals are plotted for biological specimen among PASC and non-PASC groups at days 5, 9, 14, 21 and 28 for (**a**) IL-6, (**b**) IL-10, (**c**) TNF-alpha, (**d**) MCP, (**e**) IP-10, (**f**) IFN-alpha, and (**g**) IFN-gamma, using generalized estimating equations fit with independent correlation, identity linkage, and Gaussian distribution. Covariates included sex, age, vaccination status, and SARS-CoV-2 variant. All statistics are derived from imputed data from 80 participants. Statistical significance was assessed using two-tailed tests. No statistically significant differences in estimated values were found when comparing PASC vs. non-PASC at each time point.

studies; participants had to be well enough to complete numerous visits during the first few weeks of illness. Second, the number of individuals developing PASC was relatively small, limiting our statistical power to observe clear differences even when trends were present. Third, it is possible that differences in levels of inflammation develop over time during the post-acute phase, in response to ongoing immunologic activity, and for that reason may not be present during the acute and early post-acute timepoints in this analysis.

Strengths of our study include the high degree of adherence to the biospecimen collection protocol and the collection of biospecimens in close proximity to initial symptom onset in a cohort of outpatients, who comprise the vast majority of individuals with PASC. The

inclusion of a large proportion of individuals from earlier waves of the pandemic, prior to the widespread availability of vaccination, SARS-CoV-2 treatment, or reinfection allowed us to study the natural history of this condition in the absence of these confounding factors. However, this analysis has several limitations. Due to relatively small cohort size, we were unable to evaluate whether early markers were associated with specific symptomatic phenotypes of PASC, each of which may be driven by different biology. We defined PASC as the presence of any symptom new or worse since SARS-CoV-2 infection present during the post-acute period, without requiring that the same symptom be present during the acute period; it is possible that this could result in misattribution of some unrelated symptoms to SARS-CoV-2 infection. This is mitigated somewhat by specific training for study staff to interrogate for the presence of any reported symptom prior to SARS-CoV-2 infection. We applied a relatively broad time window in which PASC could be defined (2 to 6 months), which although consistent with accepted case definitions might be subject to variability by time since infection (either waxing and waning or resolving symptoms). Future assessment of individuals during later timepoints (e.g., 6 months or beyond) following infection may be warranted. The current analysis focuses on the binary presence or absence of symptoms and does not include consideration of symptom severity or impact. Such assessment could lead to further insight into whether different viral titers and immune responses can affect the severity of symptoms rather than simply their presence or absence. Finally, although mitigated through use of multiple imputation, missing data from declined blood sample collection could be a source of error.

In summary, maximum viral RNA and N-antigen levels, burden of RNA and infectious virus shedding, and timing of antibody development may be important factors in the pathogenesis of PASC. These early biological markers may be part of a larger cascade of events during the earliest days of SARS-CoV-2 infection that warrant consideration in the larger efforts to develop a mechanistic understanding of this condition. Further research during the acute phase of SARS-CoV-2 infection, including trials that seek to alter the factors identified in this study, is required to elucidate causal mechanisms of PASC. Such efforts can eventually lead to the development of therapeutics to prevent or treat this condition.

## Methods

### Study population and procedures

This was a longitudinal cohort study enrolling individuals acutely infected with SARS-CoV-2 and their household contacts in the San Francisco Bay area between September 2020 and May 2022. Details on the protocol and procedures for enrollment have been described in detail elsewhere[35]. In brief, we enrolled participants who were recently diagnosed with nucleic acid-confirmed SARS-CoV-2 infection within 5 days of symptom onset. Molecular testing results from UCSF-affiliated sites were used by research staff to screen and recruit participants via in-person and telephone interviews. A study field team visited participants in their areas of isolation a total of 5 times during acute infection at the following timepoints: day of enrollment (ranging between day 0 and day 5 following self-reported symptom onset), and day 9, 14, 21, and 28 following symptom onset. At each study visit, blood and nasal specimens were collected. In addition, participants self-collected swabs of the anterior nares for the first 14 days post-infection, and at d17, 19, 21, and 28. Research coordinators administered phone interviews on the same day as the field visit within the acute period, then at 4 and 8 (+/- 2 months) months post-infection, using identical instruments developed in conjunction with a study of the post-acute phase[36].

### Measurements

#### Survey-based

We piloted a questionnaire containing items with key sociodemographic, medical history, symptomatology, quality-of-life,

and later vaccination information. This instrument was developed with infectious disease specialists and epidemiologists who had managed COVID-19 and PASC patients. The form was iteratively developed over time and continues to be developed during our ongoing study. Symptom items included a checklist of 32 selected symptoms followed by a free response option. The surveys were designed to assess symptom status from the time of the last interview to the current one. All symptoms reported as new, worsened or persistent since SARS-CoV-2 infection were captured; the presence of symptoms prior to acute COVID-19 illness were assessed for exclusion from later analyzes. Comorbid conditions include history of autoimmune disease, cancer, diabetes, HIV/AIDS, heart disease, hypertension, lung disease, and kidney disease.

The primary outcome of PASC was defined by the presence of any symptom reported between 2 to 6 months after the initial illness. For participants whose first follow-up visit occurred beyond 6 months after the initial illness, we assessed symptom presence within the 2 to 6-month PASC ascertainment window, aligning with definitions developed by the CDC, WHO, and, more recently, the National Academies of Science, Engineering, and Medicine[37,38].

Information on the type and number of SARS-CoV-2 vaccinations was collected at baseline and updated with each interview. Fully vaccinated participants were defined as having received a complete primary vaccine series at least 14 days prior to study enrollment.

#### Virology

As cited elsewhere[39,40], acute viral RNA levels in nasal specimens were assessed using quantitative real time reverse transcription polymerase chain reaction (PCR) to target nucleocapsid (N) and envelope (E) gene regions. Used RNAseP as a control for RNA extraction. We used 4 μL of RNA sample mixed with 5 μL 2x Luna Universal Probe One-Step Reaction Mix, 0.5 μL 20x WarmStart RT Enzyme Mix, 0.5 μL of target gene specific forward and reverse primers and probe mix (Supplemental Table 4) for each PCR reaction. Using 96-well plates, we ran PCR reactions using the following primers and probe (IDT): 5.6 μM forward/reverse each and 1.4 μM probe for N 8 μM forward/reverse each and 4 μM probe for E, 4 μM forward/reverse each and 1 μM probe for RNaseP. For each RT-qPCR plate, we ran a 10-fold serial dilution of an equal mixture of plasmids containing a full copy of nucleocapsid (N) and envelope (E) genes as the absolute standard for RNA copies calculation and primer efficiency assessment. We used the CFX Connect Real-Time PCR detection system (Biorad) with these settings: 55 °C for 10 min, 95 °C for 1 min, and then cycled 40 times at 95 °C for 10 s followed by 60 °C for 30 s. The limit of detection and the limit of quantification for the assay were established at 100 copies/mL and 1000 copies/mL, respectively. We defined a SARS-CoV-2 RNA-positive result as any level with a cycle threshold value less than or equal to 40 in the N and E gene regions. To control for the quality of self-sampling, we repeated or excluded any samples where the RNAseP cycle threshold was more than 2 standard deviations from the mean of all samples.

We assessed the presence of infectious virus through evaluation of the cytopathic effect (CPE) in Vero-hACE2-TMPRSS2 cells (BEI Resources (NR-54970)). To do so, we serially diluted 200uL of nasal specimen 1:1 with DMEM supplemented with 1x penicillin/streptomycin, then added 100uL of freshly trypsinized cells, resuspended in infection media (made as above but with 2x penicillin/streptomycin, 5ug/mL amphotericin B [Bioworld] and no puromycin) at $2.5 \times 10^5$ cells/mL. We cultured cells at 37 °C and 5% CO2 and then performed visual evaluation from day 2 to day 5. For all specimens with visible CPE, we confirmed the presence of infectious SARS-CoV-2 by RT-qPCR. To do so, 200 uL of supernatant from one well from each dilution series was mixed 1:1 with 2x RNA/DNA Shield (Zymo) for viral inactivation and RNA extraction. Two researchers. All virologic testing was performed by two individuals (MGK, MT).

Collected blood specimens were also used to measure nucleocapsid antigen (N-Ag) at all available time points using the Quanterix automated paramagnetic microbead-based immunoassay (Simoa)[41]. Compared to traditional immunoassays, this technology offers a 1000-fold greater sensitivity than the traditional immunoassay[42,43].

**Host immune factors.** Plasma biomarker measurements were performed using the automated HD-X Simoa platform. Analytes included plasma SARS-CoV-2 Spike receptor binding domain (RBD) IgG and the following inflammatory markers: IL-6, IL-10, TNF-α, IFN-α, IFN-γ, IP-10, and MCP-1.

All assays were performed according to the manufacturer's instructions and assay performance was consistent with the manufacturer's specifications.

## Statistical analysis

We describe the distribution of virologic and host immune factors at baseline and through follow-up among those with PASC compared to those without PASC. Individual-level viral shedding dynamics have been described previously[39]. Virologic factors included maximum viral RNA and infectious load, RNA decay rates (defined as the linear rate of RNA decline from maximum RNA load), the proportion with viral RNA and infectious viral shedding, and N-antigen levels.

We described the association of PCR and CPE positivity by day with the development of PASC using pooled logistic regression. Maximum viral RNA and infectious viral loads and RNA decay rates for both N- and E- targets were compared using generalized estimating equations (GEE) with an independent working correlation matrix controlling for age, sex (by self-report), vaccination status (unvaccinated versus completed primary series), and SARS-CoV-2 variant (Delta versus pre-Delta) to estimate incidence ratios, 95% confidence intervals, and p-values. Sensitivity analyzes included corrections for multiple comparisons (Supplementary Table 4).

Subgroup analyzes were conducted on participants with available blood specimens and included viral N-antigen and host immune factors, using GEEs model and controlling for the same potential confounders as the virologic analysis. Among the 5 planned specimen collection times for host immune factor measurement, 39% were not completed. Missing data were further assessed and managed using multiple imputations by chained equations assuming missingness at random[44]. We used the imputed data for the final analyzes in this report; preliminary results using non-imputed data can be found in the Supplement. The distribution and magnitude of difference of each analyte are summarized graphically. For antibody analysis we further restricted our model to unvaccinated participants to compare antibody responses. Given the nonlinear presentation of viral load and antibody results we log transformed the data for comparison.

All estimates were calculated using Stata (version 16.1; StataCorp, College Station, TX) and R including the package 'qvalue'[45].

## Human participants

The study was approved by the University of California, San Francisco (UCSF) Institutional Review Board and given a designation of public health surveillance according to federal regulations as summarized in 45 CFR 46.102(d)(1)(2). Written informed consent was obtained from all participants; for participants younger than 18 years old, written informed consent was obtained from parents/legal guardians. Participants were compensated $80 over the course of the study. This activity was reviewed by CDC and was conducted consistent with applicable federal law and CDC policy (see e.g., 45 C.F.R. part 46.102(I)(2), 21 C.F.R. part 56; 42 U.S.C. §241(d); 5 U.S.C. §552a; 44 U.S.C. §3501 et seq).

## Reporting summary

Further information on research design is available in the Nature Portfolio Reporting Summary linked to this article.

## Data availability

Numerical raw data of de-identified participant virologic and host immune factors have been deposited in Figshare (https://doi.org/10.6084/m9.figshare.26346589). Source data are provided with this paper.

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

## Acknowledgements

We are grateful to the study participants. This study was funded by the Centers for Disease Control and Prevention Broad Agency Announcement (contract 75D30120C08009). The NIH/National Institute of Allergy and Infectious Diseases also supported M.J.P. (K23AI157875) and J.D.K. (K23 AI146268). These funding sources had no role in the content of the manuscript nor the decision for publication. Additionally, we would like to acknowledge Jeremy Lambert and Quanterix Inc for providing SARS-CoV2 N-antigen and anti-RBD IgG assay kits, A. Creanga and B. Graham at NIH for providing the Vero TMPRSS2 hAce2 cells, and the study participants.

## Author contributions

S.L., M.J.P., J.D.K., S.S., and J.N.M. designed the study, supported by funding to J.D.K. S.L., M.J.P., J.D.K., M.B.H., C.M.M., J.N.M., S.S., and J.D.K. developed the methodology. S.L., M.J.P., M.C.D., K.L., J.P.R., S.A.G., J.Y.C., M.F.C., S.P., A.M., M.S., A.D., B.V., R.H., K.D., K.A., S.G.D., and J.D.K. collected and curated clinical data and biospecimens. M.T., M.G.K., A.Z., D.R.M., B.G., B.C.Y., A.C., J.W.W., C.P., and R.A. were responsible for biospecimen processing and laboratory testing. S.L., S.A.G., D.V.G., and J.D.K. performed and/or interpreted the statistical analysis. S.L., M.J.P., and J.D.K. drafted the initial manuscript. All authors edited, reviewed, and approved the final manuscript.

## Competing interests

MJP has received consulting fees from Gilead Sciences, AstraZeneca, BioVie, Apellis Pharmaceuticals, and BioNTech and research support from Aerium Therapeutics, outside the submitted work. SGD reports consulting for Enanta Pharmaceuticals and Pfizer and reports research support from Aerium Therapeutics outside the submitted work. The remaining authors declare no competing interests.

## Additional information

[1]Institute for Global Health Sciences, University of California, San Francisco (UCSF), San Francisco, CA, USA. [2]Department of Epidemiology and Biostatistics, UCSF, San Francisco, CA, USA. [3]Division of HIV, Infectious Diseases, and Global Medicine, UCSF, San Francisco, CA, USA. [4]School of Medicine, UCSF, San Francisco, CA, USA. [5]Department of Microbiology and Immunology, UCSF, San Francisco, CA, USA. [6]Departamento de Inmunologia, Instituto de Investigaciones Biomedicas, Universidad Nacional Autónoma de Mexico, Mexico City, Mexico. [7]Department of Microbiology and Immunology, Stanford, CA, USA. [8]LabCorp - Monogram Biosciences, South San Francisco, San Francisco, CA, USA. [9]Division of Respiratory Viral Pathogens, CDC, Atlanta, USA. [10]F.I. Proctor Foundation, UCSF, San Francisco, CA, USA. [11]These authors contributed equally: Scott Lu, Michael J. Peluso. [12]These authors jointly supervised this work: Sharon Saydah, J. Daniel Kelly. ✉e-mail: dan.kelly@ucsf.edu

