## [Peer Review File · Nature Communications]

Early Biological Markers of Post-Acute Sequelae of SARS-CoV-2 InfectionREVIEWER COMMENTS

Reviewer #1 (Remarks to the Author):

In the manuscript, Lu et al. delve into the development of Post-acute sequelae of SARS-CoV-2 by studying patients from the early pandemic. The authors conducted viral gene analysis during the acute infection stage and analyzed the duration of RNA and infectious viral shedding, as well as N-antigen levels in two groups of participants. Participants with PASC had higher viral load and shedding compared to Non-PASC participants. The authors extended their analysis to measure cytokine levels but found no significant difference between the two groups. While the article is generally well done, it needs improvement to strengthen the conclusion.

The paper's primary objective is to propose the use of persistent viral RNA as a biomarker for PASC. However, previous reports have found no correlation between persistent virus shedding and PASC (34358460, 33243942, 33243942). Therefore, the discrepancies in the data need to be addressed.

The authors have established a correlation between anti-RBD-IgG and PASC. Although several studies have shown that antibodies produced against cytokines and chemokines are linked to COVID-19 severity and PASC, there is still no definitive agreement on whether these responses are beneficial or harmful. In order to present it as a biomarker, more conclusive data is needed.

Figure 4 indicates no discernible difference in cytokine response between participants with PASC and those without. This contrasts with previous reports suggesting elevated levels of IL-6, TNF, and other cytokines in PASC patients, highlighting the inconsistency in findings.

Figure 3 To better illustrate the data, a graph displaying individual data points for viral shedding should be included.

The conclusion drawn in lines 37 and 195 regarding higher levels of N-antigen in PASC participants compared to non-PASC participants needs to be revised as Figure 2a does not demonstrate any significant differences between the two groups.

Including patients who are infected with different strains and have varying vaccination statuses would enhance the value of the manuscript.

Reviewer #2 (Remarks to the Author):

The authors present a novel dataset of highly sampled individuals during acute COVID-19 with a proportion of individuals eventually developing PASC. They then present several analyses and results to show which of the traits they measure are associated with the post acute symptoms. Overall, the paper reads well, but there are several instances where the manuscript would benefit clarification. Most importantly, the results have statistical shortcomings that need to be addressed, and questions that need to be at least discussed.

Major comments:

1- The population was enrolled over a very long time period (2020-2022). While this is laudable, there would have been during that time frame many different variants that would infect the study participants (as shown in table 1). It would be important to establish these variants established for each individual and to assess whether the results reported in this study are impacted by said variant.

2- What was the impact of vaccination on early markers of PASC? Does the vaccine type administered impacted the early markers differently? Were any participants vaccinated during the course of the study? This would need to be, in the best case scenario, described in the results, and at the very least, discussed in the discussion.

3- By the end of the recruiting period, effective treatments such as Paxlovid were available. Were study participants treated with any such treatment? This information needs to be collected and the impact of treatment assessed as it is likely to be a major confounder in any analyses presented here.

4- Were any biological/technical covariates used in the statistical modeling to account for unwanted variation? This is not defined in the methods. If this was not the case, it is important for the authors to provide results that shows that no confounder (sex, age, comorbidities, others) can explain the signal that they are identifying as associated with PASC, and covariates used need to be described in the methods section. For example, by looking at supplementary table 1, higher age and BMI seem to have a higher proportion of PASC individuals. This should also be formally tested and described.

5- The authors mention that participants often reported several symptoms. What was the prevalence of each symptom? And what was their co-occurrence? This important information should be presented in either figures or tables (main or supplementary).

6- Table 2 needs p values (adjusted for multi testing) associated to the Odds ratio presented. Otherwise, significant differences are not clear.

7- General statistical comment: When a statistical test is run and a p value and effect size reported, the authors do not specify the test used. This is a problem that needs to be addressed. Additionally, error bars should be added to plots when necessary.

Minor comments:

1- Did the participants have more than 1 COVID infection after the start of the study? And if so, what was the cumulative effect of infection on PASC? While this may be complex to capture, it would be a great component to discuss in the discussion.

2- Line 168-169: The authors mention that the cohort is diverse in terms of race and ethnicity. Table 1 shows otherwise, with a large majority of white individuals. This should be portrayed accurately.

3- Line 184-185: the sentence is not clear. A median of 12 what? Please clarify.

4- Line 194-195: The authors say that the RNA viral load is significantly higher in participants that develop PASC than others. There is no p value or statistic associated with

that statement either in the text or the figure. A p value and effect size need to be added to quantify any statistical statement.

5- Line 195-198: The authors state that the decay is not significantly different, and then say that it decays faster in individuals that further develop PASC. This needs to be clarified/removed as no significant difference is observed.

6- The authors report a significantly higher anti spike Igg level in individuals that do not develop PASC. How does that relate to the total Igg level for each individual? This would be interesting (although not essential) to address.

7- Figure 4 needs p values reported at relevant places.

Reviewer #3 (Remarks to the Author):

This is a very interesting study of participants in a SARS-CoV-2 cohort with frequent early sampling (both self-collected and team collected through home visits) and the relationship between early viral dynamics and subsequent post-acute persistent symptoms. Important characteristics of the cohort include recruitment during the pre-alpha variant waves (mostly ancestral and delta), the relatively young ambulatory cohort that was primarily white and Latinx. The authors conclude that with a very broad definition of PASC (at least one new/worsened/persistent symptom that occurred 2-6 months after initial symptoms), higher viral RNA burden and higher infectious virus titers in the nose, greater proportion of people with positive plasma N-antigen, delayed clearance of viral RNA from the nose, and lower plasma anti-spike IgG titers in the first 10-14 days post-symptom onset are all correlated with PASC. Though not the first paper published in an ambulatory cohort to find associations with early viral dynamics and PASC, the paper is well-presented and an important contribution due to additional variables of interest including plasma biomarkers (N antigen, anti-spike IgG, and the panel of 7 cytokines). The limitations of the data including: the size of the cohort, early variants from 2020-2021, and lack of consideration of symptom severity are all clear and enumerated in the discussion.

That said, there are a few areas that need some clarification:

1) Methods – Understanding that getting serial sampling in an ambulatory cohort can be challenging, how much of the data had to be imputed for missingness? Without imputation, were any of the associations still significant?

2) Why was such a broad definition of PASC used rather than the WHO or CDC definitions applied. Could the exact symptoms that constituted PASC be quantified—if headache, for example, was it severe enough to interrupt activities of daily living? Would suggest a sensitivity analysis or substudy to see if similar trends are seen when comparing people with severe PASC symptoms vs those without PASC.

3) Do you have the demographic characteristics of the 31 participants that had PASC compared to those who did not as well as those who ended up not being infected with SARS-CoV-2 (n=31, 73 and 61 respectively)? If run, could the longitudinal cytokines of the uninfected people be graphed with those of the infected PASC and non-PASC for comparison?

4) The Kingfisher platform is semi-quantitative and not meant to be quantitative. Do you have the data to show that you can calculate viral load (copies/ml) and the linearity in the range that you measured? The authors are making quantitative assertions using this platform's data.

Minor comments:

1) Which tool did you use for your questionnaire? Was it validated? Was it on a Likert scale?

2) How did you verify that the CPE was SARS-COV-2 related in your infectivity assay? By fluorescent specific labeling or PCR?

3) It would be good to see the data points in the figures and not just means/CI's.

REVIEWER COMMENTS

Reviewer #1 (Remarks to the Author):

In the manuscript, Lu et al. delve into the development of Post-acute sequelae of SARS-CoV-2 by studying patients from the early pandemic. The authors conducted viral gene analysis during the acute infection stage and analyzed the duration of RNA and infectious viral shedding, as well as N-antigen levels in two groups of participants. Participants with PASC had higher viral load and shedding compared to Non-PASC participants. The authors extended their analysis to measure cytokine levels but found no significant difference between the two groups. While the article is generally well done, it needs improvement to strengthen the conclusion.

The paper's primary objective is to propose the use of persistent viral RNA as a biomarker for PASC. However, previous reports have found no correlation between persistent virus shedding and PASC (34358460, 33243942, 33243942). Therefore, the discrepancies in the data need to be addressed.

RESPONSE: We thank the reviewer for bringing up these prior studies. The first (PMID 34358460) is a study from our research group in which we evaluated individuals with and without PASC symptoms 2 to 4 months post-COVID using blood and saliva samples. In that project, we did not identify a substantial proportion of individuals with detectable SARS-CoV-2 RNA in saliva at post-acute timepoints 2 to 4 months post-COVID, nor did we identify substantial differences in cellular immune responses among those with and without PASC at these timepoints. The primary finding in that study was that T-cell responses were durable in most individuals for up to 8 months. No acute-phase samples were included in that study.

The current study asks what we believe is a fundamentally different question: do the dynamics of SARS-CoV-2 during the acute phase of infection (up to 4 weeks) relate to PASC outcomes at 4 months post-COVID? We did not address the presence of post-acute viral RNA or protein in this current study. Importantly, there is no overlap between the participants in the two cohorts. We believe that the findings of each study are compatible: the viral dynamics in the acute phase matter (this study) but there is no evidence of virus persistence in saliva in the post-acute phase (PMID 34358460).

The second and third PMIDs provided by the reviewer are the same (PMID 33243942). This was a report of a single case of an individual with prolonged viral shedding who did not have PASC symptoms. This is a different clinical scenario compared to what we studied in this paper, as we focused only on viral dynamics during the first 3 weeks following infection. We therefore do not believe that these results are incompatible with the findings in the current study.

The authors have established a correlation between anti-RBD-IgG and PASC. Although several studies have shown that antibodies produced against cytokines and chemokines are linked to

COVID-19 severity and PASC, there is still no definitive agreement on whether these responses are beneficial or harmful. In order to present it as a biomarker, more conclusive data is needed.

RESPONSE: We agree additional research is needed to establish the role of anti-RBD IgG in both COVID-19 severity and PASC presence and severity. We have modified the narrative in the third paragraph of our discussion to include the following on line 279:

“Our observation of delayed anti-RBD antibody response demonstrates some evidence towards the role of host immune response in the pathologic mechanisms that lead to PASC. As our comparison of immune response was limited to a strictly unvaccinated sub-group, further assessment of the impact of interventions during the acute phase of infection is warranted.”

Figure 4 indicates no discernible difference in cytokine response between participants with PASC and those without. This contrasts with previous reports suggesting elevated levels of IL-6, TNF, and other cytokines in PASC patients, highlighting the inconsistency in findings.

RESPONSE: We believe that the reviewer is referring to a prior study from our group evaluating cytokine dynamics during the post-acute phase (see PMID 34677601). In that study, we measured using the same technology (Simoa) cytokine levels at approximately 2 and 4 months post-COVID. We found mild elevations in IL-6, TNF-alpha, and IP-10 in those with PASC compared to those who fully recovered.

The present study instead focuses on the acute phase of infection (up to 1 month), so there is no overlap in timing between the two studies. There is no clear difference at this 1-month timepoint between participants who later do/do not have PASC, but the observed levels measured at the timepoints in this study are higher (IL-6 6.00 vs 1.92, TNFa 5.22 vs 3.54, IP-10 127.13 vs 197.09) than the levels measured at timepoints in that paper (IL-6 0.95 vs 1.07, TNFa 2.34 vs 2.62, IP-10 0.60 vs 0.64), likely as a reflection of proximity in time to the onset of SARS-CoV-2 infection, which is known to be inflammatory. A potential explanation for this observation is that while the absolute levels of these markers decline with time from SARS-CoV-2 infection, differences in these markers may actually become more pronounced only as more time passes since the initial infection, once the higher inflammation during acute infection has dampened.

Given that these two studies focused on different time periods and that there was no overlap in the study participants, we do not believe that these discrepancies are incompatible. The current paper is one of the few analyses focused on relating these markers during the acute phase to PASC. Overall, we consider our findings to lend to the greater landscape of PASC pathogenesis and agree further research to confirm these observations during the acute phase is needed.

Figure 3 To better illustrate the data, a graph displaying individual data points for viral shedding should be included.

RESPONSE: We agree with the reviewers that additional data visualization on viral shedding would be valuable. Our team has published a manuscript on this previously

(PMID: 36095030). For this manuscript, we assessed the effect of PCR positivity as a proportion rather than individual data points of viral load as a more generalizable measurement of acute infection and feel this graph serves to depict the difference in PCR positivity over time between the PASC vs. non-PASC groups.

The conclusion drawn in lines 37 and 195 regarding higher levels of N-antigen in PASC participants compared to non-PASC participants needs to be revised as Figure 2a does not demonstrate any significant differences between the two groups.

RESPONSE: Thank you for making this point. We revised Figure 2a and the new 95% confidence intervals do indeed overlap. Although the baseline value of N-antigen in the blood is higher in the PASC group compared with non-PASC, the current manuscript has been revised to reflect the lack of statistical significance in the relationship between N-antigen in the blood and PASC. Furthermore, we have revised our conclusion to clarify significant and non-significant differences in our analysis.

Line 37 (now line 35) has been clarified to read:

“In comparison to those who fully recovered, those who developed PASC demonstrated significantly higher maximum levels of SARS-CoV-2 RNA, higher proportion of RNA positivity higher N antigen in the first 5 days, and lower Spike-specific IgG levels within the first 9 days of the acute phase of illness.”

Line 195 (now line 209) now reads as follows:

“Compared to those who did not develop PASC, those who developed PASC had higher maximum RNA viral load, maximum infectious viral load, and N-antigen levels (Table 2; Figure 2a). After adjustment, N-protein at day 5 ($p = 0.01$) and maximum viral loads demonstrated statistical significance (Maximum RNA N viral load: $p = 0.03$; Maximum RNA E viral load: $p = 0.04$).”

Including patients who are infected with different strains and have varying vaccination statuses would enhance the value of the manuscript.

RESPONSE: We agree with the reviewer that this would enhance our analysis and have added vaccination status and SARS-CoV-2 variant as covariates in our analysis. We have added a description to the Methods section (line 142: *“controlling for age, sex, vaccination status (unvaccinated versus completed primary series), and SARS-CoV-2 variant (Delta versus pre-Delta)”*). The fundamental conclusions have not changed.

Reviewer #2 (Remarks to the Author):

The authors present a novel dataset of highly sampled individuals during acute COVID-19 with a proportion of individuals eventually developing PASC. They then present several analyses and results to show which of the traits they measure are associated with the post acute symptoms. Overall, the paper reads well, but there are several instances where the manuscript

would benefit clarification. Most importantly, the results have statistical shortcomings that need to be addressed, and questions that need to be at least discussed.

Major comments:

1- The population was enrolled over a very long time period (2020-2022). While this is laudable, there would have been during that time frame many different variants that would infect the study participants (as shown in table 1). It would be important to establish these variants established for each individual and to assess whether the results reported in this study are impacted by said variant.

RESPONSE: We agree with the reviewer's point of controlling for variant status as a potential confounding path and in response have added this, classified as Delta and pre-Delta variant) to our analysis. We revised our adjustment set to include this variable in the analysis (line 142 of the Methods has the following added: "*controlling for age, sex, vaccination status (unvaccinated versus completed primary series), and SARS-CoV-2 variant (Delta versus pre-Delta)*"). The fundamental conclusions have not changed.

2- What was the impact of vaccination on early markers of PASC? Does the vaccine type administered impacted the early markers differently? Were any participants vaccinated during the course of the study? This would need to be, in the best case scenario, described in the results, and at the very least, discussed in the discussion.

RESPONSE: We agree vaccination status should be included and have expanded our analyses to include vaccination status at the time of sample collection, classifying participants as unvaccinated or having completed primary series (+2 weeks), also included in the edit to line 142 above. We restricted our analysis of antibody response to only unvaccinated participants (lines 157, 225, and 281), essentially examining the natural history of IgG response in development of PASC. Subsequent to the first 30-day period, a small proportion were vaccinated before assessment of PASC. We consider such an event (post-acute vaccination) to be a mediating factor and have not included that in the total minimum adjustment set as the goal of our research question is to determine the total causal effect of acute biological determinants on development of PASC.

3- By the end of the recruiting period, effective treatments such as Paxlovid were available. Were study participants treated with any such treatment? This information needs to be collected and the impact of treatment assessed as it is likely to be a major confounder in any analyses presented here.

RESPONSE: We collected data on any antiviral medication for all participants including specific medication, dates of onset, and duration of treatment. None of the participants in this analysis received Paxlovid or any antiviral medication within the period of observation. We agree that assessing antiviral use in the current era of COVID-19 treatment will be valuable in future analyses.

4- Were any biological/technical covariates used in the statistical modeling to account for unwanted variation? This is not defined in the methods. If this was not the case, it is important for the authors to provide results that shows that no confounder (sex, age, comorbidities, others) can explain the signal that they are identifying as associated with PASC, and covariates used need to be described in the methods section. For example, by looking at supplementary table 1, higher age and BMI seem to have a higher proportion of PASC individuals. This should also be formally tested and described.

RESPONSE: We appreciate this point from the reviewer and have responded by evaluating the variables needed to block potential confounding paths that could lead to bias. As a result, we have included the following adjustment set as the minimally sufficient adjustment set required to eliminate bias from confounding paths in our analysis: age, sex, vaccination status, and SARS-CoV-2 variant. This has now been clarified in the Methods section (line 142).

5- The authors mention that participants often reported several symptoms. What was the prevalence of each symptom? And what was their co-occurrence? This important information should be presented in either figures or tables (main or supplementary).

RESPONSE: We agree with the reviewer that reporting the prevalence of PASC symptoms could be valuable to the reader. Please find these estimates in the newly generated Supplementary Table 3 (Prevalence of reported symptoms among those with PASC).

6- Table 2 needs p values (adjusted for multi testing) associated to the Odds ratio presented. Otherwise, significant differences are not clear.

RESPONSE: We revised Table 2 to include p-values. We also acknowledge that there is a debate about when to adjust for multiple comparisons. As the number of comparisons increases, the argument to account for multiple comparisons with adjusted p-values becomes stronger because of the role of chance causing type I error. On the other hand, adjusted p-values can increase the probability of type II error, particularly when the number of comparisons remains relatively few. This debate is further elaborated in a publication by Ken Rothman (PMID 208137). In our manuscript, several of the biological variables being evaluated relate to viral dynamics and the collective set of results tells a story that is consistent with the biological hypothesis that acute viral dynamics may influence the development of PASC. Further, we consider the number of comparisons to be relatively few. As a result, we do not want to miss possibly important findings in this manuscript and have decided not to adjust for multiple comparisons. This is consistent with our general approach in pathogenesis-focused manuscripts like this one.

7- General statistical comment: When a statistical test is run and a p value and effect size reported, the authors do not specify the test used. This is a problem that needs to be addressed. Additionally, error bars should be added to plots when necessary.

RESPONSE: We have clarified the statistical test used in the Statistical Analysis subsection of the Methods section (line 140 now reads: *“Virologic and immunologic biomarkers were compared using generalized estimating equations with a working independence correlation matrix, identity linkage, and Gaussian distribution”*). Additionally, we have revised and added error bars to plots and included more detail, inclusive of p-values.

Minor comments:

1- Did the participants have more than 1 COVID infection after the start of the study? And if so, what was the cumulative effect of infection on PASC? While this may be complex to capture, it would be a great component to discuss in the discussion.

RESPONSE: We agree this is an important question in determining PASC status. Participants were asked about subsequent infection at each follow-up visit as part of the standard study survey. None of the participants included in this analysis reported reinfection between the time of their biospecimen collection and PASC status ascertainment.

The following has been added to line 199: *“No participants reported any reinfection at the time of PASC status ascertainment.”*, further, most of the participants in this analysis enrolled in the study before reinfections became common.

2- Line 168-169: The authors mention that the cohort is diverse in terms of race and ethnicity. Table 1 shows otherwise, with a large majority of white individuals. This should be portrayed accurately.

RESPONSE: We agree with the reviewer and have removed the word “diverse” on line 168 (line 181) as suggested by the reviewer.

3- Line 184-185: the sentence is not clear. A median of 12 what? Please clarify.

RESPONSE: We have clarified this number refers to individually reported symptoms, clarifying lines 184-185 (now 197-198) to *“a median of 12 (9 to 14) symptoms endorsed”*.

4- Line 194-195: The authors say that the RNA viral load is significantly higher in participants that develop PASC than others. There is no p value or statistic associated with that statement either in the text or the figure. A p value and effect size need to be added to quantify any statistical statement.

RESPONSE: We have revised this sentence and included the p-value in the text. We also added a column of p-values to Table 2.

5- Line 195-198: The authors state that the decay is not significantly different, and then say that it decays faster in individuals that further develop PASC. This needs to be clarified/removed as no significant difference is observed.

RESPONSE: We agree that this is confusing and have revised the line to read as follows: “Viral RNA decay rates did not significantly differ by PASC status.” to what is now line 217.

6- The authors report a significantly higher anti spike Igg level in individuals that do not develop PASC. How does that relate to the total Igg level for each individual? This would be interesting (although not essential) to address.

RESPONSE: We agree with the reviewer that this is an interesting question. Unfortunately, we did not measure total or other IgG levels in participants and are unable to include such in this analysis.

7- Figure 4 needs p values reported at relevant places.

RESPONSE: We did not observe statistically significant differences in the reported inflammatory marker levels, so we have written that in the legend and opted not to embed p-values within the Figure for simplicity. (Lines 479-480: “No statistically significant differences in estimated values were found when comparing PASC vs. non-PASC at each time point.”)

Reviewer #3 (Remarks to the Author):

This is a very interesting study of participants in a SARS-CoV-2 cohort with frequent early sampling (both self-collected and team collected through home visits) and the relationship between early viral dynamics and subsequent post-acute persistent symptoms. Important characteristics of the cohort include recruitment during the pre-alpha variant waves (mostly ancestral and delta), the relatively young ambulatory cohort that was primarily white and Latinx. The authors conclude that with a very broad definition of PASC (at least one new/worsened/persistent symptom that occurred 2-6 months after initial symptoms), higher viral RNA burden and higher infectious virus titers in the nose, greater proportion of people with positive plasma N-antigen, delayed clearance of viral RNA from the nose, and lower plasma anti-spike IgG titers in the first 10-14 days post-symptom onset are all correlated with PASC. Though not the first paper published in an ambulatory cohort to find associations with early viral dynamics and PASC, the paper is well-presented and an important contribution due to additional variables of interest including plasma biomarkers (N antigen, anti-spike IgG, and the panel of 7 cytokines). The limitations of the data including: the size of the cohort, early variants from 2020-2021, and lack of consideration of symptom severity are all clear and enumerated in the discussion.

That said, there are a few areas that need some clarification:

1) Methods – Understanding that getting serial sampling in an ambulatory cohort can be challenging, how much of the data had to be imputed for missingness? Without imputation, were any of the associations still significant?

RESPONSE: We agree these are important aspects of employing imputation and should be reported. To that end, we have added narrative on missingness and comparison to pre-imputation results.

Regarding missingness, we have added the following on lines 148-151: *“Among the 5 planned specimen collection times for host immune factor measurement, 39% were not completed. This missing data was further assessed and managed using multivariate imputation by chained equations assuming missingness at random.”*

Without imputation, we saw the same conclusions listed in the paper as well as significant relationships in IL-6, IL-10, and TNF alpha at day 21 post-symptom onset. These statistically significant associations were not seen after imputing data. We have added the following on lines 222-224: *“Prior to imputation IL-6, IL-10, and TNF-a showed statistically significant differences between those with PASC and those without PASC. After imputation, there were no significant differences observed between IL-6, IL-10, TNF-a, MCP, IP-10, IFN-a, and TFN-a (Supplemental Figure 1).”*

2) Why was such a broad definition of PASC used rather than the WHO or CDC definitions applied. Could the exact symptoms that constituted PASC be quantified—if headache, for example, was it severe enough to interrupt activities of daily living? Would suggest a sensitivity analysis or substudy to see if similar trends are seen when comparing people with severe PASC symptoms vs those without PASC.

RESPONSE: We agree with reviewers that the lack of a single unified PASC definition can be challenging from a measurement perspective. When considering PASC definitions, we considered the presence of COVID-associated symptoms and the time frame of those symptoms. On the former (new or worsened symptoms attributed to COVID-19), we consider our definition of specific symptoms to be consistent with both CDC and WHO definitions.

In response to the reviewer’s comments, we have assessed the effect of adopting the time frame indicated by the WHO, specifically looking at symptoms that present at least 3 months after acute infection. This reduced the evaluable population by 4 and did not change analysis results significantly.

Given the wide acceptance of the CDC definition, we’ve opted to keep our analysis with the current working definition in the manuscript and agree a study assessing differential PASC definitions would be interesting.

3) Do you have the demographic characteristics of the 31 participants that had PASC compared to those who did not as well as those who ended up not being infected with SARS-CoV-2 (n=31, 73 and 61 respectively)? If run, could the longitudinal cytokines of the uninfected people be graphed with those of the infected PASC and non-PASC for comparison?

RESPONSE: We agree this would lend a valuable perspective to our analysis. Unfortunately, we did not collect/test samples for those who were not identified as COVID-19 cases.

4) The Kingfisher platform is semi-quantitative and not meant to be quantitative. Do you have the data to show that you can calculate viral load (copies/ml) and the linearity in the range that you measured? The authors are making quantitative assertions using this platform's data.

RESPONSE: We appreciate the reviewer's careful consideration of our laboratory methodology. We extracted viral RNA using the Kingfisher platform and carried out RT-qPCR using methods that are now clarified in the Methods section. This includes details on the primer and probe pairs and the methods used to determine absolute copy numbers using a standard developed in-house. We validated this quantitative method by determining the assay limit of detection as detailed.

We have added the following to line 112: *"The assay limit of detection and limit of quantification were established experimentally and were 100 copies/mL and 1000 copies/mL, respectively."*

Minor comments:

1) Which tool did you use for your questionnaire? Was it validated? Was it on a Likert scale?

RESPONSE: We developed a novel questionnaire iteratively over time, beginning at the beginning of the pandemic (March 2020). The symptom-related items incorporate both specific symptom items as well as free response items; we specifically inquire about symptoms that are new or worsened since the COVID-19 diagnosis. This was supplemented with items from the Patient Health Questionnaire somatic symptom scale. In addition to the presence and absence of each symptom, we assess the severity of the symptom on a Likert scale. The questionnaire was developed in collaboration with the Long-term Impact of Infection with Novel Coronavirus study in March 2020 before Long COVID was a recognized clinical entity and has been used continually since that time in our research program. Data from the questionnaire has supported over 25 publications on Long COVID since the beginning of the pandemic. Additional details have been published previously (PMID: 35788827).

2) How did you verify that the CPE was SARS-COV-2 related in your infectivity assay? By fluorescent specific labeling or PCR?

RESPONSE: We extracted RNA from CPE assay plates and used real-time PCR targeting N to confirm samples were related to SARS-CoV-2 following the same qPCR protocol detailed in the methods but without a standard. Ct values that were 3 or more cycles below those from the nasal specimen used for the CPE assay were considered to have infectious virus. We have revised the Virology sub-section of the Methods to include our verification methods. The sentence (line 116) now reads as follows: *"Presence of*

infectious virus (infectivity) was measured via cytopathic effect (CPE) in Vero-hACE2-TMPRSS2 cells and confirmed by qPCR testing for SARS-CoV-2.”

3) It would be good to see the data points in the figures and not just means/CI's.

RESPONSE: We agree this would be an interesting presentation of the data descriptively. Given our conclusion is that there are largely no discernible differences at the level we compare, we believe means and CI's present the message most clearly.

REVIEWER COMMENTS

Reviewer #1 (Remarks to the Author):

The authors have addressed all of my concerns, but I still do not understand if a higher viral load is the cause or just a correlation to PASC. They have not shown any inflammation or metabolic changes at either an early or late stage. Therefore, I am not sure if it can be presented as a biomarker.

Reviewer #2 (Remarks to the Author):

While most major and minor comments are addressed appropriately, this reviewer has one strong criticism remaining. The authors suggest that adjusting for multi testing decreases false positive rate at the cost of fast negative rate, and use, as support, a citation from 1990 from an author that has made bold and unaccepted claims regarding the usage of p values (he also suggested that p values should not be used at all in a later publication). Furthermore, this argument is only pertaining to FWER correction such as Bonferroni correction, and not more recent methods design to control false discovery rate (this publication happened 5 years before the Benjamini-Hochberg paper describing the Benjamini-Hochberg procedure for example. More to the point, Storey's q-value method directly addresses Rothman's criticism of p-value adjustment by incorporating π_0 into the significance measure, and Storey et. al.'s later work establishing the Bayesian interpretation of q-values further obviates Rothman's 1990 arguments. Together, this emphasize the invalidity of the author's arguments regarding multi-testing, and the need to adjust for multi-testing even if results are not significant anymore for proper reproducible science. This reviewer strongly recommend that this is done in order for this paper to proceed for publication in Nature Communications.

REVIEWER COMMENTS

Reviewer #1 (Remarks to the Author):

The authors have addressed all of my concerns, but I still do not understand if a higher viral load is the cause or just a correlation to PASC. They have not shown any inflammation or metabolic changes at either an early or late stage. Therefore, I am not sure if it can be presented as a biomarker.

Response→ Our approach to assessing the early biological determinants of PASC leverages a modern epidemiological framework in the consideration of the classic threats to validity, including selection, measurement, and confounding biases. By contending with these threats, we can determine the presence of an association and build a body of evidence pointing to a potential causal inference. Furthermore, it is proposed that the early biological determinants observed in our investigation are the beginning of a biological cascade followed by viral persistence, triggering a second hit causing inflammation and metabolic changes that ultimately lead to PASC. These downstream events of the biological cascade during the post-acute stage leading to PASC are better described in the literature (Peluso et al., 2022) and not the objective of our study, whereas the research gap between the early biological determinants and PASC is much larger and the objective of our study.

Reviewer #2 (Remarks to the Author):

While most major and minor comments are addressed appropriately, this reviewer has one strong criticism remaining. The authors suggest that adjusting for multi testing decreases false positive rate at the cost of fast negative rate, and use, as support, a citation from 1990 from an author that has made bold and unaccepted claims regarding the usage of p values (he also suggested that p values should not be used at all in a later publication). Furthermore, this argument is only pertaining to FWER correction such as Bonferroni correction, and not more recent methods design to control false discovery rate (this publication happened 5 years before the Benjamini-Hochberg paper describing the Benjamini-Hochberg procedure for example. More to the point, Storey's q-value method directly addresses Rothman's criticism of p-value adjustment by incorporating π_0 into the significance measure, and Storey et. al.'s later work establishing the Bayesian interpretation of q-values further obviates Rothman's 1990 arguments. Together, this emphasize the invalidity of the author's arguments regarding multi-testing, and the need to adjust for multi-testing even if results are not significant anymore for proper reproducible science. This reviewer strongly recommend that this is done in order for this paper to proceed for publication in Nature Communications.

Response→ In accordance with the Reviewer's recommendation, we employed Storey Q-

values in evaluating virologic factors measured by GEE to strike a balance between controlling errors and maximizing the ability to detect true associations. Aware of their inability to provide information on positive predictive value of a significant result, interpretation can be further elaborated considering q-values in the context of Bayesian posterior probabilities. In analyses relating maximum RNA N viral load to PASC, for example, Storey's Q-value method estimated a p-value of 0.022 as a q-value of 0.082. We interpret a q-value of 0.082 as consistent with an association between higher maximum viral load and PASC as a correct discovery with probability of 91.8% ($1 - 0.082$). Understanding these probabilities can be helpful for the reader to consider in the context of the overall message, so we included use of q-values throughout the manuscript, fully describing and interpreting them in the Supplement (Supplemental Table 1).

REVIEWERS' COMMENTS

Reviewer #2 (Remarks to the Author):

The comments were appropriately addressed.